

# Why people select the outpatient clinic of medical centers: a nationwide analysis in Taiwan

Ming-Hwai Lin, Hsiao-Ting Chang, Tzeng-Ji Chen and Shinn-Jang Hwang

Department of Family Medicine, Taipei Veterans General Hospital, Taipei, Taiwan
School of Medicine, National Yang-Ming University, Taipei, Taiwan

## ABSTRACT

**Introduction**. In contrast to other countries, Taiwan's National Health Insurance (NHI) program allows patients to freely select the specialists and tiers of medical care facility without a referral. Some medical centers in Taiwan receive over 10,000 outpatients per day. In the NHI program, the co-payment was increased for high-tier facilities for outpatient visits in 2002, 2005, and 2017. However, the policies only mildly reduced the use of high-tier medical care facilities. The main purpose of this study was to evaluate the factors contributing to the patients' selection of the outpatient clinic of medical centers without a referral.

**Methods**. An online anonymous survey was conducted by using the Google Forms platform utilizing a self-constructed questionnaire from September to October 2018. A nationwide sample in Taiwan was recruited using convenience sampling through social media. Based on a literature review and a focus group, 20 factors that may affect the choice of the outpatient institution were constructed. The associations between items that affect the patients selection of outpatient clinics were assessed using exploratory factor analysis. Principal axis factoring was performed to identify the major factors affecting the decision. Multiple logistic regression was performed to determine which factors satisfactorily explained "visiting the outpatient clinic of the medical center for an illness without a referral."

**Results**. During the survey period, 5,060 people browsed the online survey, and 1,003 responded and completed the online questionnaire. Therefore, the response rate was 19.8%. A total of 987 valid responses was collected. Exploratory factor analysis revealed that three main factors, namely the "physician factor", "image and reputation factor", and "facility and medication factor", affected the selection of outpatient clinics. A series of logistic regressions indicated that patients who reported that hospital facilities, high-quality drugs, and diverse specialties were very important were more likely to select the outpatient clinic of a medical center (OR = 2.218, 95% CI [1.514–3.249]). Patients who reported that physician factors were very important were less likely to select a medical center (OR = 0.717, 95% CI [0.523–0.984]). Patients who were previously satisfied with their experience of the primary clinics or had a regular family doctor were less likely to choose a medical center (OR = 0.509, 95% CI -0.435–0.595] and OR = 0.676, 95% CI [0.471–0.969]).

**Conclusion**. In Taiwan, patients with good primary medical experience and regular family physicians had significantly lower rates by selecting the outpatient clinic of a medical center. The results of this study support that the key to establishing graded medical care is to prioritize the strengthening of the primary medical system.

Corresponding author
Ming-Hwai Lin,
mhlin@vghtpe.gov.tw

# INTRODUCTION

The National Health Insurance (NHI) program in Taiwan is a single-payer system founded in 1995. The NHI program comprises a hierarchy of medical care facilities consisting of four tiers: medical centers, regional hospitals, local community hospitals, and primary clinics. However, referral systems have not yet been successfully implemented.

Hierarchical medical care means that medical resources can be used the most efficiently through professional division in the medical system. In most countries, primary care physicians act as healthcare "gatekeepers" by providing initial medical interventions and referring patients to additional specialists (*Yan, Kung & Lu, 2019*). Excluding situations of major illnesses and the urgent requirement of treatment at a medical center, people should first consult a family doctor or a nearby primary clinic regarding an illness. After diagnosis and treatment, patients can be referred to other specialty clinics or hospitals.

In contrast with other countries, patients in Taiwan have full and unrestricted access to all medical care facilities. Patients in Taiwan's NHI program can freely select specialists and the tier of medical care facility directly without a referral (*Lynn et al., 2015*). The design of global budget payments and the fee for services result in patients favoring treatment at large hospitals, even for mild diseases, and medical centers are more likely to use advanced instruments and pharmaceuticals (*Kuo, Chen & Lin, 2019*; *Lee et al., 2018*). Numerous patients in Taiwan consult several physicians with different specialties and at different health care facilities and switch physicians and facilities rapidly (*Wang & Lin, 2010*). This phenomenon has been suggested as a source of inefficiency in healthcare use and has resulted in high medical expenditures and costs of outpatient visits.

Studies have reported that people in developed countries visit a doctor 5–6 times a year, whereas in Taiwan, the average frequency of visits is 13. More than 30,000 insured residents in Taiwan seek hospital inpatient and outpatient services over 100 times a year (*Lynn et al., 2015*). In certain large medical centers in northern Taiwan, the number of outpatients per day often exceeds 10,000. Furthermore, physicians frequently see over 50 patients in a morning, spending only 5 min or less for each consultation (*Wu, Majeed & Kuo, 2010*). These short consultations can cause misinformation and misunderstanding between healthcare providers and patients because of the time to build rapport. The freedom to have multiple hospital return visits results in high use of outpatient hospital visits, drug prescriptions, and other health services (*Wang & Lin, 2010*; *Yip et al., 2019*).

Excessive use of health services is a critical and persistent problem in Taiwan. To moderate these rising costs, a graded medical system was implemented in the NHI program and increased the copayment for high-tier facilities for outpatient visits in 2002, 2005, and 2017. Patients without a referral are charged an additional copayment ranging from 240 to 420 NTD (approximately 8 to 14 USD) for every visit to a high-tier medical facility. Although changes to the NHI copayment policies have mildly reduced the use of high-tier

medical care facilities, studies have indicated that the effect of medical prices on people's medical behavior is very limited (*Lee et al., 2018*). The implementation of the copayment system exerted little effect on encouraging the population visit primary clinics first (*Yang, Tsai & Tien, 2019*).

Factors affecting patients' selection of high-tier medical care facilities have not been fully identified. Cheng et al. reported that patients tend to base their judgment of hospital quality on medical equipment (*Cheng, 2015*). The main purpose of this study was to evaluate the factors contributing to the patients' selection of the outpatient clinic of medical centers without a referral. Understanding motivations underlying the public's choices would enable the implementation of a successful graded medical system in Taiwan.

## MATERIALS AND METHODS

### Study design

The present study was a web-based cross-sectional online survey. The development and reporting of the survey were performed following the guidelines of the Checklist for Reporting Results of Internet E-survey (CHERRIES) (*Eysenbach, 2004*). The checklist is available in supplementary data. The questionnaire was developed in Google Forms (https://www.google.com/forms/about/).

After the initial tests and revision of the questionnaire were completed, and a nationwide sample in Taiwan was recruited using convenience sampling through an online anonymous survey from September 3 to October 31, 2018. The questionnaire was administered to various community groups by using the snowball sampling method. To maximize public outreach, the survey was promoted on various social media platforms, such as Facebook, Line, and the most popular bulletin board system (https://facebook.com/; https://linecorp.com/; https://www.ptt.cc/index.bbs.html). Interested citizens were invited to complete the questionnaire and respondents were asked to invite their friends to participate in the survey and fill out the questionnaire.

The link to the survey was available for 8 weeks. All participants were invited to complete an anonymous self-administered online questionnaire, which required approximately 10 min to complete. Informed consent was requested from all participants on the first page of the questionnaire. Only participants who were at least 20 years old and were able to read Chinese fluently were given access. No rewards were provided to participants. A deduplication protocol was applied to identify multiple submissions and preserve data integrity, including cross-validation of the eligibility criteria of key variables and discrepancies in key data (*Bowen et al., 2008*).

This study was approved by the Institutional Review Board of Taipei Veterans General Hospital (2017-07-009AC), and the study was conducted following the guidelines of the Helsinki declaration of 2013.

### Questionnaire design

A questionnaire was developed because no similar questionnaires related to the selection of outpatient clinics are available. The questionnaire was finalized after experts were invited to review and revise. A literature search was performed for publications that discussed the

factors affecting the selection of outpatient clinics. Search terms used were "health care seeking behavior", "hospital outpatient clinics", and a combination of the two. Based on factors identified in the literature search, two family physicians, three outpatient nurses, and five volunteers were invited to participate in the focus group. The main topic was "What are the important factors in the selection of an outpatient clinic by a patient". Opinions provided by the experts were used as a reference for the questionnaire.

Based on a literature review and the opinions of the focus group, factors that related to the selection of outpatient clinic were proposed and included in the questionnaire. The main dependent variable of this study was "preferred choice of outpatient clinics when you are ill", and the independent variables were assessed using the following question: "Please indicate the importance of each of the following factors in your selection of an outpatient clinic when you were ill?" A total of 20 factors affecting the choice of the outpatient institution was included. All respondents were asked to rate the importance of the 20 factors in the selection of an outpatient institution when they were ill on a 5-point Likert scale ranging from 1 = not at all important to 5 = very important.

At the end of the questionnaire, respondents were asked to provide demographic information and information on past experiences during outpatient visits at different hospital levels, attitudes towards copayment, and whether they have a regular family physician. The questionnaire was developed based on a literature review and the opinions of the focus group to ensure content validity. Five senior researchers with subject matter expertise were invited to revise the questionnaire and perform repeated testing of the questionnaire. The content was rated by five experts with an average content validity index of 86.0%. The questions were refined after feedback and finalized into an online survey.

At the beginning of the study, the questionnaire was pretested in 20 patients to determine if the content was appropriate and to ascertain whether the content was understandable. The internal consistency reliability test was used for reliability analysis. Cronbach's alpha of the questionnaire was 0.895, which is satisfactory.

## Statistical analysis

All data were stored on a secure server and backed up on a local hard disk to ensure the security of the data. Only the researcher could access these materials. The data were primarily evaluated by Dr. Lin, Ming-Hwai. The survey data were extracted into Excel (Microsoft Corp), and statistical analyses were performed using the Statistical Package for the Social Sciences (SPSS, version 20.0; SPSS Inc., Chicago, IL, USA).

Descriptive statistics were used to present the results for patient hospital choices. Independent samples $t$-tests and Chi-square tests were adopted to examine the association between respondents' demographic characteristics and their outpatient preference. The normality of the collected data was analyzed using the Kolmogorov–Smirnov test. The data followed a normal distribution; thus, comparisons among the three groups were performed using analysis of variance (ANOVA). A $p$ value of $< 0.05$ (two-tailed) was considered statistically significant.

The associations between items that affect the patients' choice of outpatient clinics were assessed using exploratory factor analysis. Measures of sample adequacy, such as Kaiser-Meyer-Olkin (0.868) and Bartlett's Test of Sphericity (significance < 0.001), indicated that factor analysis could be applied. Principal axis factoring was performed to identify the major factors by using a correlation matrix and oblimin rotation. The number of principal components to be extracted was determined by examining the eigenvalues (> 1). Loadings of over 0.5 were used to interpret components in the study. The number of domains was reduced to three and named "physician factor", "image and reputation factor", and "facility and medication factor". Internal consistency was demonstrated, with the factors' Cronbach's $\alpha$ coefficients ranging from 0.792 to 0.905. These three factors accounted for 61.7% of the total variance of the variables.

Multiple logistic regressions were performed to determine factors that satisfactorily explained the dependent variable "visiting the outpatient clinic of the medical center for an illness without a referral." The adjusted odds ratios (ORs) with 95% confidence intervals (CIs) for predicting "visit to the outpatient clinic of a medical center for an illness" were computed. In model 1, the association of age, gender and personal experience of primary clinics was tested. The physician factor, image and reputation, and facility and medication factors were included in model 2 to test the associations beyond the personal factors. The other variables were included in model 3 to test the association of sociodemographic factors, in addition to the aforementioned factors.

## RESULTS

During the survey period, 5,060 people browsed the online survey, and 1,003 responded and completed the online questionnaires. Therefore, the response rate was 19.8%. We excluded 16 participants because of duplication (the same age, occupation, and answer options). Table 1 provides a comparison of the demographic characteristics of the patients who favor different institutions for outpatient visits.

The mean age of the respondents was 43.6 years (SD, minimum, and maximum were 10.6, 19, and 85 years, respectively). Men accounted for 43.8% and women accounted for 56.2% of the 987 respondents included; 509 (51.6%) respondents favored visiting a primary clinic, 308 (31.2%) favored visiting the general hospital, and 170 (17.2%) favored visiting the medical center without a referral. Table 1 provides a comparison of demographic characteristics and preferred institutions for outpatient visits. Gender, marital status, and education level were not statistically related to the choice of outpatient visits. In univariate analysis, the choice of medical treatment facility was statistically related to income ($p = 0.026$). Patients with a monthly income of NTD 50,001–70,000 favored outpatient clinics of medical centers. People living in urban areas accounted for 65.8% of respondents. A larger number of people living in urban areas favored medical centers than patients living in other areas ($p < 0.001$). Approximately 51.5% of the respondents had regular family doctors. Significantly more patients who favor primary clinics for outpatient visits had regular family doctors than patients who prefer medical centers (61.9% vs 41.2%, $p < 0.001$). Approximately 67.6% of the respondents were satisfied with their previous

**Table 1 Demographic characteristics and preferred institution for outpatient visits (N = 987).**

| | Total | Preferred institution for outpatient visit | | | p value |
| | | Primary clinic | General hospital | Medical center | |
|---|---|---|---|---|---|
| | n = 987 | n = 509 | n = 308 | n = 170 | |
| | n (%) | n (%) | n (%) | n (%) | |
| age (mean, SD) | 43.6 (10.6) | 41.7 (10.7) | 43.6 (10.3) | 49.6 (8.8) | |
| sex: male | 432 (43.8) | 221 (43.4) | 138 (44.8) | 73 (42.9) | 0.902 |
| educational level | | | | | 0.927 |
| tertiary or below | 149 (15.1) | 76 (14.9) | 48 (15.6) | 25 (14.7) | |
| university | 647 (65.6) | 338 (66.4) | 201 (65.3) | 108 (63.5) | |
| postgraduate | 191 (19.4) | 95 (18.7) | 59 (19.2) | 37 (21.8) | |
| marriage | | | | | 0.193 |
| married | 644 (65.2) | 328 (64.4) | 195 (63.3) | 121 (71.2) | |
| others | 343 (34.8) | 181 (35.6) | 113 (36.7) | 49 (28.8) | |
| income | | | | | 0.026 |
| NTD <15000 | 168 (17.0) | 90 (17.7) | 50 (16.2) | 28 (16.5) | |
| NTD 15001–30000 | 130 (13.2) | 70 (13.8) | 37 (12.0) | 23 (13.5) | |
| NTD 30001–50000 | 346 (35.1) | 180 (35.4) | 120 (39.0) | 46 (27.1) | |
| NTD 50001–70000 | 176 (17.8) | 74 (14.6) | 57 (18.5) | 45 (26.5) | |
| NTD >70000 | 167 (16.9) | 95 (18.7) | 44 (14.3) | 28 (16.5) | |
| area | | | | | <0.001 |
| urban | 649 (65.8) | 337 (66.2) | 179 (58.1) | 133 (78.2) | |
| suburban/rural | 338 (34.2) | 172 (33.8) | 129 (41.9) | 37 (21.8) | |
| residency | | | | | 0.059 |
| northern | 662 (67.1) | 335 (65.8) | 199 (64.6) | 128 (75.3) | |
| middle | 115 (11.7) | 59 (11.6) | 40 (13.0) | 16 (9.4) | |
| southern | 163 (16.5) | 96 (18.9) | 48 (15.6) | 19 (11.2) | |
| east/archipelagos | 47 (4.8) | 19 (3.7) | 21 (6.8) | 7 (4.1) | |
| have a regular family physician | 508 (51.5) | 315 (61.9) | 123 (39.9) | 70 (41.2) | <0.001 |
| satisfied with the experience of the primary clinic | 667 (67.6) | 383 (75.2) | 194 (63.0) | 90 (52.9) | <0.001 |

medical experience in primary care. Furthermore, patients who favored primary clinics for outpatient visits exhibited significantly higher satisfaction rates than patients who favored medical centers (75.2% vs 52.9%, $p < 0.001$).

Table 2 summarizes the average rating of respondents on the importance of each factor when selecting an outpatient facility and their preferred outpatient institution. "Physicians were highly reputable", "physicians explained in detail", and "physicians have a good medical practice" were the rated most important factors to consider when selecting the outpatient institution. The low copayment was the least important factor for outpatient medical choice among all patients (Likert scale rating of $3.08 \pm 1.16$).

In univariate analysis, six factors were significantly more important among the respondents who chose to visit a medical center ($p < 0.001$). These factors were "physicians are highly reputable", "physicians have a good medical practice", "the institution has

**Table 2  Association between the average rating of respondents to each factor when selecting an outpatient facility and their preferred outpatient institution.**

| factors considered when selecting an outpatient facility | Total | Preferred institution for outpatient visit | | | p value |
| --- | --- | --- | --- | --- | --- |
| | | Primary clinic | General hospital | Medical center | |
| | $n = 987$ | $n = 509$ | $n = 308$ | $n = 170$ | |
| | average rating of respondents | | | | |
| physicians are highly reputable | 4.65 ± 0.71 | 4.66 ± 0.69 | 4.55 ± 0.78 | 4.81 ± 0.58 | 0.001*** |
| physicians explained in detail | 4.57 ± 0.75 | 4.58 ± 0.74 | 4.49 ± 0.80 | 4.68 ± 0.69 | 0.027* |
| physicians have a good medical practice | 4.47 ± 0.80 | 4.40 ± 0.82 | 4.46 ± 0.77 | 4.66 ± 0.72 | 0.001*** |
| consider the severity of the disease | 4.37 ± 0.91 | 4.34 ± 0.94 | 4.36 ± 0.85 | 4.48 ± 0.94 | 0.235 |
| the institution has advanced equipment | 4.35 ± 0.86 | 4.25 ± 0.86 | 4.34 ± 0.85 | 4.65 ± 0.79 | <0.001*** |
| the institution has high-quality drugs | 4.34 ± 0.92 | 4.28 ± 0.93 | 4.28 ± 0.95 | 4.62 ± 0.75 | <0.001*** |
| physicians are not in a hurry | 4.30 ± 0.87 | 4.32 ± 0.88 | 4.22 ± 0.89 | 4.40 ± 0.81 | 0.071 |
| physicians are gracious and kind | 4.25 ± 0.85 | 4.25 ± 0.85 | 4.22 ± 0.85 | 4.30 ± 0.86 | 0.645 |
| have good medical experience | 4.24 ± 0.79 | 4.25 ± 0.79 | 4.20 ± 0.79 | 4.29 ± 0.80 | 0.414 |
| the institution has friendly staff | 4.15 ± 0.96 | 4.12 ± 1.00 | 4.14 ± 0.91 | 4.22 ± 0.94 | 0.500 |
| the institution has convenient transportation | 4.13 ± 0.96 | 4.11 ± 0.95 | 4.13 ± 0.94 | 4.18 ± 1.03 | 0.722 |
| the institution has diverse specialties | 4.09 ± 0.99 | 3.97 ± 1.06 | 4.12 ± 0.90 | 4.39 ± 0.87 | <0.001*** |
| waiting time is not too long | 3.90 ± 0.93 | 3.91 ± 0.93 | 3.94 ± 0.86 | 3.78 ± 1.02 | 0.171 |
| the institution was recommended by friends or relatives | 3.55 ± 0.99 | 3.54 ± 1.01 | 3.50 ± 0.89 | 3.71 ± 1.05 | 0.074 |
| institutions with a good reputation | 3.53 ± 1.03 | 3.46 ± 1.03 | 3.45 ± 0.99 | 3.88 ± 1.05 | <0.001*** |
| the visibility of medical institutions is high | 3.43 ± 1.06 | 3.39 ± 1.07 | 3.41 ± 0.97 | 3.62 ± 1.18 | 0.042* |
| willing to prescribe for chronic diseases | 3.40 ± 1.16 | 3.36 ± 1.16 | 3.40 ± 1.16 | 3.50 ± 1.16 | 0.381 |
| physicians are famous | 3.32 ± 0.98 | 3.25 ± 0.97 | 3.31 ± 0.94 | 3.52 ± 1.03 | 0.007** |
| physicians with a good reputation | 3.29 ± 0.91 | 3.23 ± 0.90 | 3.26 ± 0.91 | 3.50 ± 0.89 | 0.003** |
| low copayment | 3.08 ± 1.16 | 3.07 ± 1.14 | 3.15 ± 1.17 | 3.00 ± 1.19 | 0.394 |

**Notes.**
*** $p < 0.001$.
** $p < 0.01$.
* $p < 0.05$.

advanced equipment", "the institution has high-quality drugs", "the institution has diverse specialties", and "the institutions has a good reputation". In this study, we conducted exploratory factor analysis to understand the potential common characteristics among factors and clarify the influencing factors. We used principal component analysis to extract data using a correlation matrix and oblimin rotation method. We removed six items because of cross-loading or because the factor load was too low ($<0.4$). Factors with eigenvalues greater than 1, cumulative percentages of variance explained above 71.2%, KMO value reaching of 0.868, and p value less then 0.001 were excluded. Three main factors were retained in the final extraction (Table 3), namely "physician factor", "image and reputation factor", and "facility and medication factor". We subsequently converted the scores to three factors into a multivariable analysis model.

Table 4 illustrates three models of logistic regression for predicting "visits to the outpatient clinic of the medical center for an illness." Age was a crucial predictor in all the models. The likelihood of choosing to visit a medical center when ill increased by

**Table 3** Exploratory factor analysis loads and variance percentages for factors considered when selecting an outpatient facility.

| Factor items | Factors loads | | |
|---|---|---|---|
| | Factor I: physician factor | Factor II: image & reputation | Factor III: facility & medication |
| physicians explained in detail | 0.922 | | |
| physicians are highly reputable | 0.855 | | |
| physicians are not in a hurry | 0.851 | | |
| physicians are gracious and kind | 0.780 | | |
| the ability of the physician is well known | 0.488 | | |
| physicians with a good reputation | | 0.851 | |
| physicians are famous | | 0.747 | |
| institutions with a good reputation | | 0.656 | |
| the visibility of medical institutions | | 0.545 | |
| the institution has advanced equipment | | | −0.817 |
| drug quality is trustworthy | | | −0.781 |
| diverse specialty | | | −0.741 |
| sum of squared loading (eigenvalue) | 5.239 | 1.515 | 0.653 |
| percentage of variance explained (%) | 43.659 | 12.627 | 5.438 |
| cumulative percentage of variance explained (%) | 43.659 | 56.286 | 61.724 |
| Cronbach's alpha | 0.905 | 0.840 | 0.792 |

**Notes.**

Kaiser–Meyer–Olkin (KMO): 0.868 Bartlett sphericity tests ($P < 0.001$).

Six factors were removed because the factor load was too low ($< 0.4$) or because of cross-loading. The removed factors were "consider the severity of the disease", "institution has convenient transportation", "reasonable waiting time", "institution was recommended by friends or relatives", "willing to prescribe for chronic diseases", and "low copayment".

2.7%–3.1% for every additional year of age (95% CI [1.4%–4.3%]) when other variables were controlled.

In Model 1, when age, gender, "have a regular family physician," and "consider that copayment is important" were adjusted, patients who were previously satisfied with the medical experience of primary clinics had a 0.5 lower likelihood of visiting the outpatient clinic of a medical center for an illness (95% CI [0.429–0.584]).

Model 2 was then also adjusted for the extracted factors I to III, which revealed that patients who reported that hospital facilities, high-quality drugs, and diverse specialties were very important had a 2.218 higher likelihood of selecting the outpatient clinic of the medical center (OR = 2.218, 95% CI [1.514–3.249]). Patients who were previously satisfied with the medical experience of primary clinics had a 0.509 lower likelihood of choosing a medical center to visit when ill (95% CI [0.435–0.595]). Patients who rated copayment as important were 0.525 times as likely to select a medical center to visit when ill (95% CI [0.354–0.781]). People with a regular family doctor were 0.676 times less likely to select a medical center (95% CI [0.471–0.969]). Patients who rated physician factors as very important were less likely to select an outpatient clinic in a medical center (OR = 0.717,

**Table 4  Results of the logistic regression for predicting "visit to an outpatient clinic of the medical center for an illness".**

| Variables | MODEL 1 | | | MODEL 2 | | | MODEL 3 | | |
|---|---|---|---|---|---|---|---|---|---|
| | Exp(B) | 95% CI of OR | | Exp(B) | 95% CI of OR | | Exp(B) | 95% CI of OR | |
| age | 1.031*** | 1.019 | 1.043 | 1.028*** | 1.016 | 1.041 | 1.027*** | 1.014 | 1.041 |
| male | 0.821 | 0.582 | 1.159 | 0.817 | 0.575 | 1.162 | 0.786 | 0.544 | 1.134 |
| past experience in primary clinics | 0.500*** | 0.429 | 0.584 | 0.509*** | 0.435 | 0.595 | 0.557*** | 0.467 | 0.663 |
| have regular family physician | 0.694* | 0.489 | 0.986 | 0.676* | 0.471 | 0.969 | 0.659* | 0.457 | 0.952 |
| consider copayment is important | 0.643* | 0.441 | 0.938 | 0.525*** | 0.354 | 0.781 | 0.547** | 0.365 | 0.818 |
| factor I: physician factor | | | | 0.717* | 0.523 | 0.984 | 0.896 | 0.705 | 1.137 |
| factor II: image and reputation | | | | 1.257 | 0.975 | 1.621 | 1.289* | 1.042 | 1.593 |
| factor III: facility and medication | | | | 2.218*** | 1.514 | 3.249 | 1.802*** | 1.392 | 2.332 |
| lived in an urban area | | | | | | | 1.286 | 0.844 | 1.957 |
| lived area:northern Taiwan | | | | | | | | | |
| middle Taiwan | | | | | | | 0.763 | 0.416 | 1.398 |
| southern Taiwan | | | | | | | 0.572* | 0.330 | 0.989 |
| eastern Taiwan | | | | | | | 1.220 | 0.484 | 3.073 |
| education: high school | | | | | | | | | |
| college | | | | | | | 0.742 | 0.465 | 1.184 |
| postgraduate | | | | | | | 0.753 | 0.412 | 1.374 |
| income:NTD ≤ 30000 | | | | | | | | | |
| NTD 30001–50000 | | | | | | | 0.692 | 0.433 | 1.107 |
| NTD >50000 | | | | | | | 1.064 | 0.670 | 1.689 |
| −2log likelihood | | 854.516 | | | 812.212 | | | 798.631 | |
| Model χ² | | 513.757 (df = 5) *** | | | 556.061 (df = 8) *** | | | 569.642 (df = 16) *** | |
| Step χ² | | 513.757 (df = 5) *** | | | 42.304 (df = 3) *** | | | 13.581 (df = 8) p = 0.093 | |
| Nagelkerke R2 | | 0.541 | | | 0.574 | | | 0.585 | |
| percentage of correctly classifying the outcome | | 82.0% | | | 82.9% | | | 82.7% | |

**Notes.**
*** ***$p \leq 0.001$.
** **$p \leq 0.01$.
** $^P \leq 0.05$.

95% CI [0.523–0.984]). The gender of the patient and the image and reputation of the hospital and physicians were not significantly related to inpatient hospital choice.

In Model 3, when the possible sociodemographic confounding variables were added, the step Wald chi-square statistic was insignificant (Wald chi-square difference = 13.581, $df = 8$, $p = 0.093$). The residential area, income, and education level did not appear to be related to the selection of an outpatient clinic. Therefore, we decided to adopt model 2 as the result of our analysis.

## DISCUSSION

Several factors significantly affected the selection of a medical center, including older age, the physician factors, advanced equipment, high-quality drugs, past experience in primary clinics and the copayment. Most of the Taiwanese population agree with the principle of a hierarchical medical system and a medical referral system. However, many people still

disagree with changes to their health care-seeking choices because of policy promotion (*Yan, Kung & Lu, 2019*). A survey determined that age, residence, education, and monthly family income were significantly related to inpatient hospital choice (*Kamra, Singh & De, 2016*). Some results were consistent with ours. However, in our study, income did not have an obvious effect on outpatient choice. This may be because of the exemption for low-income people in Taiwan's health insurance. Low-income residents do not pay any component when visiting a medical center without a referral (*Yang, Tsai & Tien, 2019*).

Family physicians were introduced over 20 years ago in Taiwan. However, only 51.5% of the respondents had regular family doctors. In this study, patients with regular family doctors, who were satisfied with their medical experience in primary care, who rated the physician factor as important, and who rated copayment as important, were less likely to select a medical center when ill. These results indicated that the implementing a family physician system, whereby the public generally has a trusted family doctor, would help reduce the number of patients electing to go directly to the medical centers without a referral.

Gender, marital status, and education level did not affect the choice of outpatient visits. The univariate analysis indicated that the choice of the outpatient institution was only slightly related to income levels, and income levels were not related to the outpatient choice, after controlling for other variables in regression analysis. Low copayment was the least important factor for outpatient medical choice among all patients. This result may be caused by the low copayment amount in Taiwan's NHI system. Furthermore, in the NHI program, most of the cost of medical treatment is waived for low-income households and catastrophic illness patients in Taiwan. Thus, the financial burden is rarely a consideration in the patients' choice of outpatient institution (*Chen & Fan, 2015*). The current copayment of outpatient medicines is a fixed fee, and the out of pocket maximum is only NTD\$200 (approximately USD\$6.7). Although the NHI copayment reforms had mildly reduced the probability that patients with minor ailments would choose to visit high-tier medical facilities, several studies have indicated that the effect of medical prices on people's medical behavior is limited.

In the present research, a similar phenomenon was also observed. Low copayment had the lowest average rating on the Likert scale when considering the importance of outpatient medical choices among all patients. Changes to the health insurance system (e.g., changing the copayment to a fixed-rate coinsurance) may be the only method to eliminate unnecessary testing and medical waste (*Victor et al., 2018*).

Ideally, every older adult should have trusted primary care physicians who can provide outpatient services. However, in this study, older people had a greater likelihood to visit the medical center for outpatient visits. In 2012, Liu et al. indicated that the different health profiles of elderly people significantly affected the likelihood of use and expenditure on health care services. The high comorbidity group tended to use more ambulatory care services, and the frail group had higher health care expenditures (*Liu, Tian & Yao, 2012*). Our research results did not accord with these findings. Further research is needed to understand whether the primary clinics in Taiwan satisfy the needs of elderly people.

This study has several limitations that may affect the findings. First, participants were recruited over the Internet because of the web-based survey design, thus the low response rate warrants further exploration. Although the online survey represents a wide age range and geographic distribution, the sample was younger and more highly educated than the general public (*Tengilimoglu et al., 2017*). Hsieh et al. determined that Internet use in Taiwan was significantly associated with more outpatient clinic visits among people with chronic diseases in Taiwan (*Hsieh et al., 2016*); therefore , caution should be exercised when generalizing these results. Second, the variance explained by the logistic regression model suggests that other significant factors may determine outpatient clinic decisions (*Cheng, 2015*; *Yip et al., 2019*).

Despite these limitations, this study is the first to investigate how the public chooses outpatient institutions in Taiwan. Further research should explore the influencing factors among the older group.

## CONCLUSIONS

A good primary medical experience and a regular family physician significantly reduces people's likelihood of visiting the medical center without a referral. The results of this study support that the key to establishing graded medical care is prioritizing the strengthening of the primary medical system.

### Funding

This study was supported by grants from the Ministry of Science and Technology (MOST 106-2314-B-075 -032 -MY3) and Taipei Veterans General Hospital (V107C-095). The funders had no role in study design, data collection and analysis, decision to publish, or preparation of the manuscript.

### Grant Disclosures

The following grant information was disclosed by the authors:
Ministry of Science and Technology: MOST 106-2314-B-075 -032 -MY3.
Taipei Veterans General Hospital: V107C-095.

### Competing Interests

The authors declare there are no competing interests.

### Author Contributions

- Ming-Hwai Lin conceived and designed the experiments, performed the experiments, analyzed the data, prepared figures and/or tables, authored or reviewed drafts of the paper, and approved the final draft.
- Hsiao-Ting Chang analyzed the data, prepared figures and/or tables, authored or reviewed drafts of the paper, and approved the final draft.
- Tzeng-Ji Chen and Shinn-Jang Hwang conceived and designed the experiments, authored or reviewed drafts of the paper, and approved the final draft.

## Human Ethics

The following information was supplied relating to ethical approvals (i.e., approving body and any reference numbers):

The Institutional Review Board of Taipei Veterans General Hospital approved this research (2017-07-009AC).

## Data Availability

The raw data and codebook are available as Supplementary Files.

## Supplemental Information

Supplemental information for this article can be found online at http://dx.doi.org/10.7717/peerj.9829#supplemental-information.

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
