# Peer review of "Why people select the outpatient clinic of medical centers: a nationwide analysis in Taiwan"

_PeerJ, doi:10.7717/peerj.9829_

## Round 0.1 · original submission · Major Revisions

In addition to the Reviewers comments, I am attaching a pdf file of the paper with my detailed comments. I have some concerns that need to be addressed:

1. The questionnaire development method needs explanation in Methods. This includes a description of the focus group and which previous studies were used to develop the questionnaire.

2. I did not understand the rationale behind the use of 3 models in Results. They are not explained in Methods, they include a mix of the variables derived from the PCA and the individual questions and some background variables are included in one model and the others in another model. This part needs revision: explanation in Methods, clarification of the rationale, which factors are included and why.

3. The dichotomization of the 5-point Likert scale responses to important vs not leads to loss of details. An alternative is to treat the scores as quantitative variables and report their means then compare the means among the 3 groups in table 2 using ANOVA/ similar tests.

4. In various sections (Discussion and otherwise), some parts do not fit and should be shifted to the sections where they belong or removed altogether. Details are included in my comments.

5. The Discussion should include comparison to previous studies.

6. Some references need to be updated- they are very old. Pls check the attached pdf.

Reviewer 1 ·

Basic reporting

The reference style needs to be checked and matched with the Journal's style.

Experimental design

Methods is not described with sufficient detail & information to replicate.

Validity of the findings

The low response rate inherent in Internet-based survey could affect the validity of the findings.

Additional comments

The manuscript reported the results of a nationwide Internet-based survey on the factors associated with patients’ visiting the outpatient clinic of the medical center for an illness without a referral. Multiple logistic regression analysis showed that age, past experience in primary clinics, consider copayment is important, have regular family physician, living area, and two factors (image and reputation & facility and medication) obtained from an exploratory factor analysis were significantly associated with the outcome variable. The conclusion appeared to be supported by the results, but a number of clarifications need to be made, particularly regarding the study methodology.

1. Line 103: Please provide more details on how the survey was advertised on the three social media platforms. Is there a link to a study survey website, such as Google Form?

2. Line 109: Was the deduplication protocol based on only identical age, occupation, and answer options? Were IPs checked for duplication? Is there a survey report generated by the survey website that allows one to roughly identify the geographical distribution of the survey respondents? If so, this will add support to the “nationwide” coverage of the survey.

3. Line 119: Please provide more information on the focus group, including the demographic characteristics of its member and the main questions used.

4. Line 120: “expert validity” should be referred to as “content validity”. Was Content Validity Index (CVI) measured? If so, this should be used to replace the sentence (line 129) “it exhibited a satisfactory level of content validity”.

5. Line 142 and 187: A p-value of less than .001 should be presented as < .001 rather than .000.

6. Line 143: While the results may not be different, principal axis factoring rather than principal components method should be used for the factor analysis because it appears that the aim of the authors were trying to assess a set of latent variables that cannot be directly measured with a single variable.

7. Line 145 and 190: “Multiple logistic regression” rather than “multivariate logistic regression” is the correct term to use here.

8. Line 156: The age distribution should be described. In addition to the mean and SD, please provide the minimum and maximum age of the participants.

9. Line 191 and Table 4: The authors used hierarchical (nested) regression models, which should be first described in the Methods section. Please explain why this approach rather than stepwise regression approach was used.

10. Line 241: The low response rate should also be mentioned in the limitations.

11. References: They are a number of formatting errors, such as missing page number (Line 287), and the use abbreviated journal title (PeerJ style is journal title in full form) (Line 273, 275, 277, etc.).

12. Table 4: The columns for beta and standard error of beta should be deleted and replaced by columns showing p values and 95% confidence intervals for the odds ratios.

13. Table 4: Income was represented by a categorical variable with five levels in Table 1. However, it was represented by a single variable “income degree”. Are the five levels treated as a continuous variable? Unless there are special reasons for this, it should be analyzed as a categorical variable.

14. Table 4: The models should be compared formally with likelihood-ratio Chi-square test. It is not clear whether Model 3 is significantly “better” than Model 2.

Reviewer 2 ·

Basic reporting

- The manuscript is presented clearly and unambiguously, using professional English. The text is easy to understand.

- Bibliographic references follow the APA style, according to the instructions for the journal's authors. However, errors have been detected in the citation in the text. For example, "Chen et al. 2006” on line 65. According to the regulations, up to 3 authors must be determined, while if you have 4 or more authors, you must specify the first followed by et al. Authors are recommended to review APA citation style and make relevant changes to the text.
Despite the fact that the references are current and fit the theme of the study, the introduction needs more bibliographic support. Suggest that you quote other authors on lines 61-62, 72-74, and 86-92 to provide more justification for your study:
"In most countries, primary care physicians act as 'gatekeepers' to health care by providing initial medical interventions and referring patients to additional specialists." You must comply with this statement with bibliographic references.
"Studies have reported that people in developed countries visit a doctor 5 to 6 times a year, while in Taiwan, the average frequency of visits is 13." You must comply with this statement with bibliographic references.
"Although changes in NHI co-pay policies have slightly reduced the use of high-level healthcare facilities, studies have indicated that the effect of medical prices on people's medical behavior is very limited (Lee et al . 2018).
Factors affecting patient selection in high-level healthcare settings have not been fully identified. Cheng et al. reported that patients tend to base their judgment on the quality of the hospital on the medical team (Cheng, 2015). "Please note that this explanation seems somewhat contradictory when mentioning the influence of medical prices and the quality of hospitals in the choice by the patients, once they mention that the factors that completely affect this question have not been completely identified This precise fragment is scientifically clarified and supported.

- The structure of the manuscript and tables is understandable and adjusted to the specific ones. We suggest that in the materials and methods part the authors elaborate a sub-section on ethical and legal aspects for lines 112-113 “No rewards were given to the participants. This study was approved by the Institutional Review Board of the Taipei General Veterans Hospital (2017-07-009AC). "In addition, we recommend that you expand the information on the legal guidelines that the authors followed to protect the confidentiality of the data.

- The data presented is consistent and fully presented to interpret the results. The tables are clear and expand the information without repeating the data.

Experimental design

- The research is within the objectives and scope of the journal.

- We suggest that the authors restate the objective of the abstract study as "The main purpose of this study was to explore the factors contributing to patients' selection of the outpatient clinic of medical centers without a referral." We recommend that the authors change the verb “explore” (mostly used in qualitative studies) to “identify or evaluate”, since they use a Likert scale questionnaire and not interviews or open questions.

- The authors must write a short justification of the study together with the objective at the end of the introduction.

- They should briefly identify the knowledge gap being investigated and statements should be made about how the study contributes to filling that gap.

- Also, keep in mind that the hypothesis proposed must be clearly defined, relevant and significant.

- We suggest that the authors expand the information in relation to the elaboration of the questionnaire that allows its correct interpretation, as well as its reproducibility.

- Also, it is recommended that the authors express more clearly based on which criteria the group of experts were selected and how they were recruited for the identification and review of the factors expressed in the questionnaire.

- In general, keep in mind that the methods must be described with enough information so that another researcher can reproduce them.

Validity of the findings

- It is important to note that the questionnaire was previously tested on 20 patients and was valid and reliable to measure the factors that influence the choice of the medical center.

- Despite the fact that the data seems reliable and statistically sound, the authors are encouraged to provide information on who was responsible for managing and performing the analysis and interpretation of the data, and how this procedure was carried out.

- We recommend the authors to move the first paragraph of the conclusions to the discussion section, since due to its content it is more appropriate.

- The conclusions must respond to the stated objective and should be limited only to those supported by the results expressed in the manuscript.

Additional comments

We want to thank the authors for considering the journal PeerJ - The Journal of Life and Environmental Sciences for the publication of their manuscript entitled “Why do people in Taiwan select the outpatient clinic of the medical center? A nationwide analysis ”.
The topic addressed is important for the development of graduated medical care, and even more so if it is a first foray into it to give priority to the primary medical system.
Despite the importance of the topic covered and the aspects previously discussed, some aspects of the manuscript require review by the authors for clarity and reproducibility.

Other aspects that we suggest to the authors are the following:

- We recommend that the authors add the study design in the abstract.

- Regarding the keywords, we suggest that they change "national health insurance" to "national health programs" and "survey" to "health care surveys" to improve the presence of the manuscript in the databases.

- In addition, we recommend reviewing the rest of the keywords and, if possible, replacing them with appropriate MeSH terms according to the topic of the study.

- Finally, it is important that the authors mention in the manuscript the reporting guidelines they have followed according to their type of study, with the aim of improving the quality and transparency of their research. If you have questions, you can consult the instructions for the authors of the magazine or https://www.equator-network.org/

---

## Round 0.2 · Minor Revisions

The MS is greatly improved. Some concerns remain:

1- There is a need to revise the entire MS for English.

2- The regression used is not hierarchial logistic, it is a series of logistic regression models. Please check (https://amstat.tandfonline.com/doi/abs/10.1080/01621459.1985.10478148#.XxBHaigzbic) for clarification and correct throughout.

3- There is some confusion about the interpretation of the models in table 4. If the 3 models are to remain, the interpretation of the results needs to follow the models so that the variables included in model 2 or 3 are not mentioned when model 1 is interpreted. The authors also did not interpret the improved fit from model I to III. The final model III should form the basis of the Discussion since it is the fully adjusted model. Please check detailed comments in the attached file.

Reviewer 1 ·

Basic reporting

No further comments.

Experimental design

No further comments.

Validity of the findings

No further comments.

Additional comments

The authors have satisfactorily responded to all my comments and made the necessary changes to the manuscript.

Reviewer 2 ·

Basic reporting

No comment.

Experimental design

No comment.

Validity of the findings

No comment.

Additional comments

Dear authors,

We thank you for the new submission of your manuscript 47877v2 entitled "Reasons Why People Select the Outpatient Clinic of the Medical Center: A National Analysis in Taiwan."

Changes made after reviewers' comments have improved the clarity and reproducibility of their manuscript, as well as their understanding by readers.

---

## Round 0.3 · accepted · Accept

The MS is greatly improved. My concerns are addressed. Congratulations